# Effects of Austenitization Temperature and Pre-Deformation on CCT Diagrams of 23MnNiCrMo5-3 Steel

**DOI:** 10.3390/ma13225116

**Published:** 2020-11-13

**Authors:** Ivo Schindler, Rostislav Kawulok, Petr Opěla, Petr Kawulok, Stanislav Rusz, Jaroslav Sojka, Michal Sauer, Horymír Navrátil, Lukáš Pindor

**Affiliations:** 1Faculty of Materials Science and Technology, VŠB—Technical University of Ostrava, 17. listopadu 2172/15, 70800 Ostrava, Czech Republic; rostislav.kawulok@vsb.cz (R.K.); petr.opela@vsb.cz (P.O.); petr.kawulok@vsb.cz (P.K.); stanislav.rusz2@vsb.cz (S.R.); jaroslav.sojka@vsb.cz (J.S.); michal.sauer@vsb.cz (M.S.); horymir.navratil@vsb.cz (H.N.); 2Technology and Research, TŘINECKÉ ŽELEZÁRNY, a. s., Průmyslová 1000, 73961 Třinec, Czech Republic; lukas.pindor@trz.cz

**Keywords:** low-alloy steel, austenitization temperature, plastic deformation, phase transformations, CCT diagram, dynamic recrystallization

## Abstract

The combined effect of deformation temperature and strain value on the continuous cooling transformation (CCT) diagram of low-alloy steel with 0.23% C, 1.17% Mn, 0.79% Ni, 0.44% Cr, and 0.22% Mo was studied. The deformation temperature (identical to the austenitization temperature) was in the range suitable for the wire rolling mill. The applied compressive deformation corresponded to the true strain values in an unusually wide range. Based on the dilatometric tests and metallographic analyses, a total of five different CCT diagrams were constructed. Pre-deformation corresponding to the true strain of 0.35 or even 1.0 had no clear effect on the austenite decomposition kinetics at the austenitization temperature of 880 °C. During the long-lasting cooling, recrystallization and probably coarsening of the new austenitic grains occurred, which almost eliminated the influence of pre-deformation on the temperatures of the diffusion-controlled phase transformations. Decreasing the deformation temperature to 830 °C led to the significant acceleration of the austenite → ferrite and austenite → pearlite transformations due to the applied strain of 1.0 only in the region of the cooling rate between 3 and 35 °C·s^−1^. The kinetics of the bainitic or martensitic transformation remained practically unaffected by the pre-deformation. The acceleration of the diffusion-controlled phase transformations resulted from the formation of an austenitic microstructure with a mean grain size of about 4 µm. As the analysis of the stress–strain curves showed, the grain refinement was carried out by dynamic and metadynamic recrystallization. At low cooling rates, the effect of plastic deformation on the kinetics of phase transformations was indistinct.

## 1. Introduction

The individual phase transformations can be effectively controlled by the cooling rate of the rolled or forged steel products. This significantly contributes to achieving the desired combinations of mechanical properties of the material. The kinetics of austenite decomposition is not only fundamentally influenced by the chemical composition of the steel but also by the parameters of the structure entering the given phase transformation. The prior microstructure depends on the austenitization conditions, the parameters of the pre-deformation, and the cooling time [1,2,3,4,5,6]. The continuous cooling transformation (CCT) diagrams show the temperatures of the phase transformations, and the individual decomposition products of austenite depend on the cooling at different rates. They are always constructed for the specific austenitization parameters and initial microstructure. The deformation continuous cooling transformation (DCCT) diagrams include the effect of pre-deformation, characterized mainly by the strain value [7,8,9,10,11].

The very complex influence of chemical composition on the kinetics of austenite decomposition has been studied in detail in many works. For example, the combined effect of manganese, silicon, nickel and vanadium [12], niobium, copper, silicon and molybdenum [13], carbon, manganese, silicon, chromium, molybdenum and vanadium [14], or niobium alone as the microalloying element content has been investigated in different steels [15,16].

From the point of view of the influence of the prior structure, the size of austenitic grains is a crucial parameter (both the initial and secondary grain size, which can also be influenced by the austenitization conditions) [17,18]. The cause of the coarsening of the prior structure is a high heating temperature, a long stay at the austenitization temperature, or a long time interval between austenitization (or deformation at this temperature) and the phase transformation itself; this applies in particular to the very low cooling rates. On the contrary, dynamic recrystallization induced by deformation at a suitable combination of temperature and strain rate can be used to refine the austenitic grains [19,20,21]. The influence of the austenitic grain size is significant, especially where transformations take place with a diffusion mechanism, which preferably uses the boundaries of the original grains for the nucleation of the new phase components. In the case of smaller austenitic grains, the matrix contains a higher density of grain boundaries and thus more nucleation sites, which mainly accelerates the ferritic transformation [18,19,20,21,22,23,24]. The fine-grained austenitic microstructure also leads to a reduction in the critical deformation for a ferrite formation by dynamic strain-induced transformation (DSIT) and to a relatively homogeneous size distribution of ferritic grains. Conversely, in the case of a coarse-grained prior structure, the newly formed ferritic grains are primarily located along the boundaries of the austenitic grains [25,26,27,28]. However, this conclusion is only valid in the case of allotriomorphic ferrite. With the increasing size of the prior austenitic grains, intragranular nucleation of ferrite indirectly favors (e.g., at inclusions located inside the austenitic grains), leading to the formation of idiomorphic ferrite instead of allotriomorphic ferrite [29]. The austenitic grain size also plays an important role in the kinetics of pearlitic transformation, as the most favorable conditions for a pearlite nucleation are in the high-energy locations. These places in the homogeneous austenite are mainly the grain boundaries, where the greatest accumulation of crystal lattice defects is. In the case of coarse-grained steels, the rate of austenite → ferrite transformation is lower, because the density of sites favorable to nucleation is low [22,30,31,32].

The coarse austenitic grains support the martensitic transformation by increasing the temperature of its start and finish. The austenitic grain size also impacts the morphology of martensite, because the coarse-grained structure affects the size of martensitic plates or needles [20,33,34].

The major effect of the previous plastic deformation should be the displacement of the phase transformations to higher temperatures and shorter times. This is caused by the increased diffusivity of elements in the deformed austenite and a higher density of the favored nucleation places for the diffusional transformation products. Such places are found particularly in the shear bands and grain boundaries with high density of dislocations [35,36,37].

The accelerating effect of plastic deformation on ferritic or pearlitic transformation has been confirmed by many researchers—see, e.g., [14,22,38,39,40,41]. This phenomenon is used in practice, for example, in achieving ultrafine-grained structures through strain-induced ferrite transformation (DSIT) of austenite to ferrite [42]. The strain-induced ferrite transformation (DIFT) has been identified as a very effective mechanism contributing to the grain refinement of the transformation-induced plasticity (TRIP) steel at just below the austenite → ferrite transformation temperature [43].

The impact of plastic deformation on the bainitic transformation varies depending on the strain value and especially on the chemical composition of the steel. There are two opposing phenomena in deformed austenite. The nucleation of the new phase component is accelerated, because the nuclei first form in the deformation bands as narrow ferritic particles lined with carbides. At the same time, however, the growth of nuclei by the shear mechanism is delayed. The growth rate of nuclei can also be affected by the deformation-induced precipitation of carbides, which act as barriers to the movement of the phase interface. However, if the plastic deformation takes place even during the bainitic transformation itself, this structure-forming process is significantly accelerated [44,45,46,47,48].

As for the martensitic transformation, the previous plastic deformation usually has a slight decelerating effect. During the deformation of austenite, a dense dislocation network is formed, which slows down the phase interface movement. Despite the large number of nuclei, the portion of the new phase component tends to be smaller than in the case of the transformation of undeformed austenite, especially at higher cooling rates. Sometimes, however, the opposite phenomenon occurs when the accumulated lattice defects initiate the formation of martensite, and it begins to form at relatively higher temperatures [49,50,51].

The influence of the strain value on the kinetics of phase transformations is ambiguous. Varying the strain value from 5 to 25% had no influence on the phase transformations in manganese–nickel and manganese low-alloy steels but modified the microstructure after cooling due to the formation of coarser austenite grains under deformation of the “critical” strain value [52]. It depends considerably on the deformation temperature and thus on whether the already recrystallized austenitic or deformed grains enter the phase transformation. This is evidenced by the comparison of the CCT and DCCT diagrams constructed for the 34CrMo4, 42CrMo4, 52CrMo4, 51CrV4, and 34NiMo6 steels at a strain value of 0, 30, and 60% [22]. Due to the high temperature of plastic deformation (i.e., 1200 °C), the individual phase transformations were not affected directly by the deformation strengthening but only indirectly by the size of the recrystallized austenitic grain. Therefore, due to the pre-deformation, only slight changes in the curves corresponding to the start and finish of the phase transformations were observed.

## 2. Experimental Material and Methods

The aim of the experimental work was to investigate the combined effect of the deformation temperature and strain value on the continuous cooling transformation diagram of steel low-alloyed with manganese, nickel, chromium, and molybdenum. The deformation temperature (identical to the austenitization temperature) should be in the range suitable for the particular continuous wire rolling mill. The applied plastic deformation intentionally corresponded to the true strain values in an unusually wide range (from 0 to 1.0) to reflect the intensive accumulation of strengthening during multi-pass wire rolling in the finishing blocks.

The 23MnNiCrMo5-3 steel is currently one of the most required materials for the production of chains, in terms of price–quality ratio. After cooling from the finish-rolling temperature, it is suitable to achieve the highest possible bainite content in its microstructure. According to operational experience, bainite appears to be the most suitable structural component for annealing, which precedes the actual production of chains. This steel can be used for the welded round link chains and various components for chain hoists, chain conveyors in the mining industry, etc. The final properties of these products are then affected by quenching and tempering. The chemical composition of this low-alloy steel, tested in the hot rolled state, is presented in Table 1. The steel was melted and cast in TŘINECKÉ ŽELEZÁRNY a. s. (Třinec, Czech Republic).

For the purpose of the experiment, the simple cylindrical samples with a diameter of 6 mm and length of 86 mm were prepared from the wire with a diameter of 10 mm. The dilatometric tests were performed with the use of the Model 39,112 Scanning Non-Contact Optical Dilatometer and Extensometer System with Green LED Technology of the Gleeble 3800 Hot Deformation Simulator (Dynamic Systems Inc., Poestenkill, NY, USA). Radial components of the strain of the sample can be measured with a repeatability of ±0.3 µm by a frequency of up to 2400 Hz.

The first step was to determine the transformation temperature *A_c3_* (i.e., the temperature at which the ferrite is completely transformed into austenite by the heating process). The sample was heated with a rate of 5 °C·s^−1^ to a temperature of 500 °C; then, the heating rate was slowed down to 0.167 °C·s^−1^ to reach the maximum temperature of 1000 °C. Evaluation of the measured data was carried out using the semi-automatic CCT software (Dynamic Systems Inc., Poestenkill, NY, USA), which applies the tangential method in combination with the derivative of the dilatation curve to determine the transformation temperatures. The result of this test is presented in Figure 1.

Based on the experimentally obtained value *A_c3_* = 801 °C and the finish-rolling temperature commonly used on the continuous wire mill in TŘINECKÉ ŽELEZÁRNY a. s., the austenitization temperature *T_A_* = 880 °C was used in the first stage of the dilatometric experiments. This value corresponds to the requirements of DIN 17115 German standard for the quenching temperature of the studied steel, which is 870 to 890 °C. The samples were resistively heated at a rate of 10 °C·s^−1^ in the measured zone to the austenitization temperature and after a duration of 600 s, they were deformed by uniaxial compression at a strain rate of 1 s^−1^ to the value of the true strain *e* = 0.35 or 1.0 (for construction of the DCCT diagrams), or no deformation was applied (i.e., *e* = 0; for construction of the CCT diagram) before the controlled cooling to almost room temperature. Constant nominal cooling rates in the range of 0.2–50 °C·s^−1^ were achievable using the samples described above. To achieve cooling rates of 100 or 200 °C·s^−1^, the samples had to have a special hollow-head shape for high-speed cooling realized by air nozzles. Unfortunately, such a sample does not allow any deformation in principle before dilatometric measurement in the course of cooling. Figure 2 shows the shape of the central parts of the simple dilatometric samples after various deformations.

The dilatometric results were verified by metallographic analyses and HV30 hardness measurement; these tests were performed on a cross section, taken in the middle of the length of the heated part of the samples. Samples intended for the light-microscopy analysis were prepared by using mechanical grinding and polishing. The microstructure was revealed via etching with the 4% Nital solution (a mixture of nitric acid and ethanol) and observed on the Olympus GX51 inverted metallurgical microscope (Olympus Corporation, Tokyo, Japan). In the case of visualization of the prior austenitic grain boundaries in the selected quenched samples, a solution of 50 g of picric acid and 5 g of ferric chloride in 100 mL of distilled water was used as an etchant. The quenched samples were pre-heated to 40 °C, and the etching time was chosen differently based on the partial results. A linear intercept technique for measuring the average size of austenitic grains by software QuickPHOTO Industrial 3.2 (PROMICRA, s.r.o., Prague, Czech Republic) was used, and only the grains with clearly drawn boundaries were taken into account. At least 217 grains were thus measured for each sample.

After evaluating the results of the first stage of the dilatometric tests, the experiments were repeated in the second stage for a lower austenitization temperature of 830 °C. This value is based on the operating limits of the relevant rolling mill and is particularly interesting in terms of the possible refinement of the prior austenitic grains. In this case, only strains *e* = 0 or *e* = 1.0 were used. In total, two CCT diagrams and three DCCT diagrams were compiled for the investigated steel.

Stress–strain curves were experimentally determined for the strain rate of 1 s^−1^ and deformation temperatures of 880 °C and 830 °C. Uniaxial compression tests were performed on a Gleeble 3800 simulator. Cylindrical specimens with a diameter of 8 mm and a height of 12 mm were heated in the same mode as in the case of dilatometric tests. The critical strain values of *e_c_* required to initiate dynamic recrystallization were calculated from the obtained flow curves according to the procedure described and applied, e.g., in [53,54,55,56]. In the beginning, the first derivative of the stress- strain curve in its up-to-peak region with respect to the true strain has to be gained. Secondly, the obtained derivative (specifically the dependency of the logarithm of strain-hardening rate ln *θ* versus true strain *e*) is described by means of the third-degree polynomial. The inflection point, i.e., the result of the second derivative of this polynomial with respect to the true strain put into the equality with zero, then corresponds with the required value of strain *e_c_*—see Figure 3 for an example. The analysis of the flow curves was performed using data smoothing and numerical derivative in the OriginPro 9 software (OriginLab Corporation, Northampton, MA, USA).

## 3. Results and Discussion

### 3.1. CCT and DCCT Diagrams

The graphs in Figure 4, Figure 5, Figure 6, Figure 7 and Figure 8 show the temperatures of the individual phase transformations (black points) for different experimental conditions; these values were determined based on the analysis of dilatometric curves. The manually interpolated curves form the boundaries of areas characterized by the existence of a particular phase component—ferrite (F), pearlite (P), bainite (B), and martensite (M).

The chemical composition of the investigated steel ensures the presence of bainite in the structure already after cooling at a rate of 0.2 °C·s^−1^. To induce a martensitic transformation, cooling at a rate of approx. 2–3 °C·s^−1^ is always sufficient. The decomposition of austenite to ferrite and pearlite is significantly affected only by deformation at low austenitization temperature.

### 3.2. Phase Composition of Samples after Dilatometry

The results of metallographic analyses are summarized in Table 2. The fractions of the structural components are arranged from the largest (left) to the smallest (right). The meaning of the parentheses is as follows: (very small amount), [rare, trace amount]. The data in Table 2 are documented by the microstructure of selected samples after dilatometry (see Figure 9, Figure 10, Figure 11, Figure 12 and Figure 13).

It should be noted that martensite was found in all samples, at least in trace amounts, even after cooling at the lowest rate. This occurrence of martensite was so small that it could not be manifested in the analysis of the dilatation curves. The minor local occurrence of martensite in microsegregations (probably of manganese and chromium [57]) was documented in the photographs (see Figure 9a for an example) but was not included in Table 2. In case of the cooling rate minimization and quasi-equilibrium conditions’ achievement in the dilatometry of the investigated steel, the fraction of ferrite and perlite would further increase at the expense of the bainite and martensite content. However, due to the existence of microsegregations, some trace occurrence of martensite cannot be excluded even under these conditions.

For comparison, a metallographic analysis of the initial hot-rolled structure of the investigated steel was also performed (see Figure 14). Due to its phase composition, this microstructure is close to structures obtained after dilatometric testing from the lower austenitization temperature and using the cooling rate of about 3 °C·s^−1^. The hot-rolled microstructure appears to be somewhat coarser.

### 3.3. Hardness Influenced by the Cooling Rate

Table 3 presents the hardness values measured for selected samples after dilatometry. The HV30 hardness in the range of about 225 to 495 was achieved. As the cooling rate increases, the total fracture of bainite and martensite in the structure increases, leading to steady growth in hardness. After cooling at a rate of 50 °C·s^−1^ and higher, the structure consists almost exclusively of martensite and the hardness remains practically stable.

### 3.4. Comparison of the Results

The comparative graph in Figure 15 shows that, for 23MnNiCrMo5-3 steel, it is not possible to unambiguously determine the effect of deformation on the kinetics of phase transformations after high-temperature austenitization. Both the bainitic and martensitic regions remain almost intact by the pre-deformation. The only significant deviation is the course of the perlite-start curve after deformation *e* = 0.35.

Such behavior is surprising and inconsistent with the results of many other studies on the effect of plastic deformation on the continuous cooling transformation diagrams [1,2,3,5,6,10,14,22,38,39,40,41,58]. However, less unambiguous data can be found comparing the CCT and DCCT diagrams for some steels. The effect of austenitization temperature (940 °C or 1000 °C) as well as the pre-deformation (true strain of 0 or 0.35) was quite insignificant in the case of high-carbon steel with 0.73% C [37]. Relatively low strain values of up to 25% had almost no effect in manganese–nickel and manganese low-alloy steels [52]. For the HSLA steel microalloyed with niobium and vanadium, the CCT and DCCT (strain *e* = 0.35) diagrams differed only slightly after austenitization at 900 °C [7]. At cooling rates below about 4 °C·s^−1^, the ferrite-start temperatures were even lower in the case of pre-deformation. This deceleration of austenite → ferrite transformation at the slow cooling rates could have been caused by the static recrystallization of deformed austenite, followed by a certain grain growth. A similar result was presented in article [1] for steel containing 0.28% C, 1.41% Mn, 0.26% Cr, 0.22% Mo, 0.027% Nb, 0.028% Ti, 0.019% V, and 0.003% B; the austenitizatin temperature was 885 °C and true strain *e* = 0.69. The pre-strain *e* = 0.2 after austenitization at 900 °C did not influence the ferrite-start transformation temperatures at all, and the ferrite-finish transformation temperatures increased only at cooling rates above 5 °C·s^−1^ [8].

In the case of low-temperature austenitization, the results for the 23MnNiCrMo5-3 steel already correspond to the theoretical assumptions—see Figure 16. The plastic deformation almost does not affect the austenite → bainite or austenite → martensite transformation, but it fundamentally expands the ferrite and pearlite regions towards the higher cooling rates. At low cooling rates (approx. below 2 °C·s^−1^), the effect of deformation on the ferrite-start or pearlite-start temperature is more or less not manifested.

Qualitatively similar results were obtained when testing bainitic steel containing 0.04% C, 1.0% Mn, 1.0% Cr, and 0.065% Nb after austenitization at 880 °C [59]. The pre-deformation with a total strain of 0.6 and a strain rate of 1 s^−1^ had a great effect on the structural constituents. In the case of the CCT diagram, the microstructure was a mixture of bainitic ferrite and granular bainite over a wide range of the cooling rates. Plastic deformation of austenite changed the microstructure to fully polygonal ferrite at all the investigated cooling rates, apart from the appearance of a very small fraction of pearlite at cooling rates ranging from 2 to 20 °C·s^−1^. The acceleration of the phase transformation austenite → ferrite due to plastic deformation was very effective in this case.

The graph in Figure 17 also corresponds well to the comparative Figure 15 and Figure 16. Considering the common scatter of the measured values, it can be stated that after high-temperature austenitization, the hardness is not affected by the previous strain value—see the trend indicated by the solid curve. In the medium cooling rate range, the hardness of the samples austenitized at 830 °C is relatively lower. This is particularly evident in the case of the pre-deformation *e* = 1, primarily due to the increased content of soft ferrite (see the dashed line).

A definite explanation of the causes of these phenomena is not easy, because we do not always have evidence of the nature of the austenitic structure entering the relevant phase transformation. This mainly concerns the low cooling rates, leading to the decomposition of austenite into ferrite and pearlite. Conversely, after higher cooling rates, it is relatively easier to obtain information about the prior austenitic grains by etching the as-quenched material. The visualization of such grain boundaries is documented by examples in Figure 18.

After deformation and cooling, the austenitic grains were always more or less equiaxed but with considerably different sizes. This indicates an uneven course of recrystallization and probably also the growth of some recrystallized grains.

The prior austenitic grain data for samples cooled at a rate 10 °C·s^−1^ are compared in Figure 19. Confidence intervals (see the red vertical line segments) were estimated for the significance level *α* = 0.05.

Reducing the austenitization temperature by 50 °C decreased the austenitic grain size after heating by almost half. The plastic deformation of *e* = 1.0 resulted in grain refinement of about 50% at an austenitization temperature of 880 °C and 45% at an austenitization temperature of 830 °C, respectively. However, these changes in the prior structure significantly affected the kinetics of phase transformations only in the case of low-temperature austenitization. Thus, it is not only the absolute grain size that could be decisive but probably also the type of softening processes taking place during the cooling of the deformed austenite. The graph in Figure 20 plots parts of the hot flow curves around the stress peak corresponding to both austenitization temperatures and strain rate of 1 s^−1^. From the calculated values of *e_c_*, it is clear that at both used deformation temperatures the strain *e* = 0.35 is sufficient to start the dynamic recrystallization. Consequently, a combination of metadynamic and static recrystallization can be expected during cooling in this case. The strain *e* = 1.0 corresponds to a steady state for both temperatures, and only metadynamic recrystallization should occur during cooling. Thus, the post-dynamic softening mechanisms did not differ significantly after the intense plastic deformation performed at a temperature of 830 °C or 880 °C. The key will be the role of cooling time, of course longer in the case of the higher austenitization and deformation temperature. At low cooling rates, these times are prolonged, which allows the more perfect softening of the deformed structure and possibly also coarsening of the austenitic grains before the actual phase transformation. The decomposition of austenite is markedly different only in the case of a single combination of experimental conditions (*T_A_* = 830 °C and *e* = 1.0), when there is a significant shift of the ferrite-start and pearlite-start curves towards the shorter times (see Figure 15 and Figure 16). In this case, the austenitic grain size of 4.0 µm is almost 40% smaller than in the test performed at *T_A_* = 880 °C and *e* = 1.0 (i.e., 4.0 µm vs. 6.5 µm—see Figure 19). This is true for the area of medium cooling rates (close to 10 °C·s^−1^), where the differences between the CCT and DCCT diagrams in Figure 16 are most pronounced. During slow cooling (about below 2 °C·s^−1^), the differences in austenitic grain size are probably blurred, as evidenced by the agreement of the ferrite-start and pearlite-start temperatures in the transformation diagrams corresponding to the low austenitization temperature.

Generally, the austenitization temperature can have a major effect on the kinetics of phase transformations during the cooling of the deformed structure. The prior grain size and the state of precipitates in HSLA steel are important [7], leading to a significant acceleration of the austenite → ferrite transformation at cooling rates above about 4 °C·s^−1^ after austenitization at 900 °C compared to preheating at 1280 °C. The effect of the very high temperature of austenitization (e.g., 1200 °C vs. 885 °C in [1]) is strong and results in the shift of the C-curves for diffusion controlled transformations towards the longer times. Experiments performed on 34CrMo4 steel showed a minor effect of a high austenitization and deformation temperature of 1200 °C on the DCCT diagram in comparison with the CCT diagram [22]. A compressive deformation of 30 or 60% only slightly changed the transformation behavior except for a rise in the martensite-start temperature.

It should be emphasized that the results obtained for 23MnNiCrMo5-3 steel are valid to true strain rate *ė* = 1 s^−1^. Changing the parameters of dilatometric tests could lead to rather different results. The problem is probably not the direct influence of the strain rate on the kinetics of phase transformations [19,27] but rather influencing the type of the interconnected softening processes during plastic deformation and during subsequent cooling. It may be important whether dynamic and thus metadynamic recrystallization can be induced, or whether static recrystallization will dominate [60,61,62]. The critical strain values of *e_c_* required to initiate dynamic recrystallization in various alloys is a function of the temperature-compensated strain rate (i.e., the Zener–Hollomon parameter *Z* (s^−1^)) according to the Sellars model [63,64,65,66,67,68,69]:(1)$$e_{c}=A\cdot Z^{B},$$
(2)$$Z=\dot{e}\cdot \exp\left(\frac{Q}{R\cdot T}\right),$$
where *T* (K) is temperature, *R* = 8.314 J·mol^−1^·K^−1^ is the gas constant, and *A* and *B* are the material constants. As the strain rate increases, the critical strain *e_c_* will grow, and the probability of at least partial metadynamic recrystallization during cooling will decrease. In addition, the dependence according to Equation (2) can be complicated by varying the size of prior austenitic grains [70].

## 4. Conclusions

Using dilatometric tests and metallographic analyses, a total of five different continuous cooling transformation diagrams were made for 23MnNiCrMo5-3 steel. Pre-deformation corresponding to the compressive true strain of 0.35 or even 1.0 had no clear effect on the austenite decomposition kinetics at the austenitization temperature of 880 °C. The reason is too great a difference between this high deformation temperature and the *A_c3_* temperature determined to be 801 °C. During the long-lasting cooling, recrystallization and probably coarsening of the new austenitic grains occurred, which practically eliminated the influence of plastic deformation on the temperatures of the diffusion-controlled phase transformations.

Decreasing the austenitization and deformation temperature to 830 °C has already led to the expected results regarding the effect of plastic deformation on the individual phase transformations in steels. In the region of the cooling rate approximately between 3 °C·s^−1^ and 35 °C·s^−1^, there was a significant acceleration of the austenite → ferrite and austenite → pearlite transformations due to the applied strain of 1.0. A shift of the curves of ferrite-start and pearlite-start towards the shorter times is evident. On the contrary, the kinetics of the bainitic or martensitic transformation remained almost unaffected by the pre-deformation. The acceleration of the diffusion-controlled phase transformations results from the formation of a fine-grained austenitic microstructure with a mean grain size of about 4 µm. As the analysis of the stress–strain curves showed, the grain refinement was performed by dynamic and metadynamic recrystallization in this case. At low cooling rates, the effect of plastic deformation on the kinetics of phase transformations was more or less blurred again.

The obtained results showed the key importance of the austenitization or finish-rolling temperature in combination with the cooling rate for the structure-forming processes taking place in the investigated steel. The DCCT diagrams enable us to optimize the cooling rate of the continuously rolled wire and thus obtain a suitable microstructure with a sufficient fraction of bainite. The CCT diagram corresponding to the austenitization temperature of 880 °C is important for determining the cooling rate, which ensures a fully martensitic microstructure after the final hardening of the high-strength chains.

## Figures and Tables

**Figure 1 materials-13-05116-f001:**
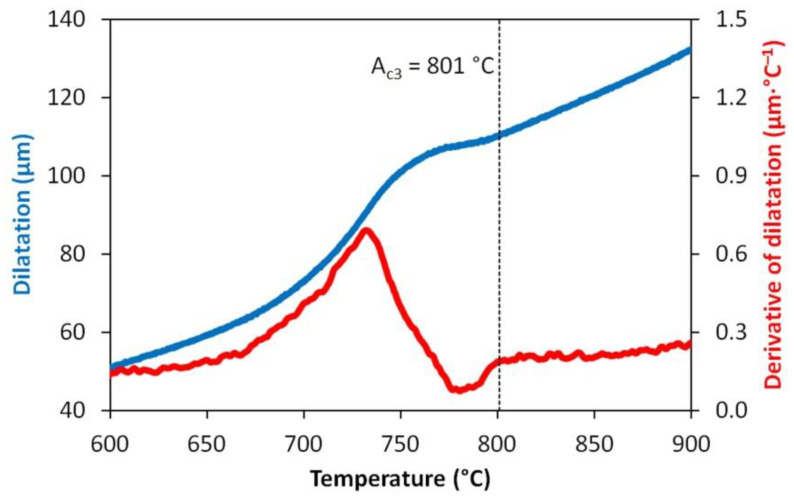
Determination of the transformation temperature *A_c3_* for the tested steel.

**Figure 2 materials-13-05116-f002:**
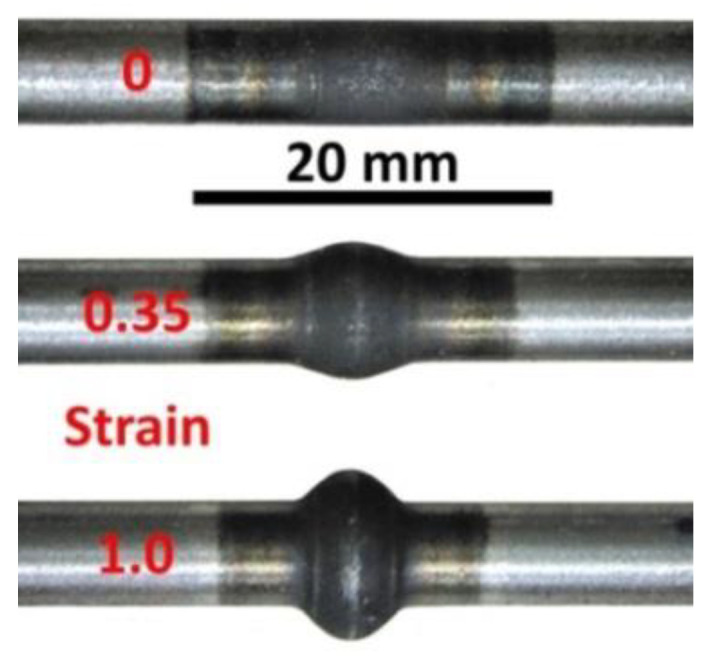
The shape of the samples after dilatometric testing—details of the heated central part.

**Figure 3 materials-13-05116-f003:**
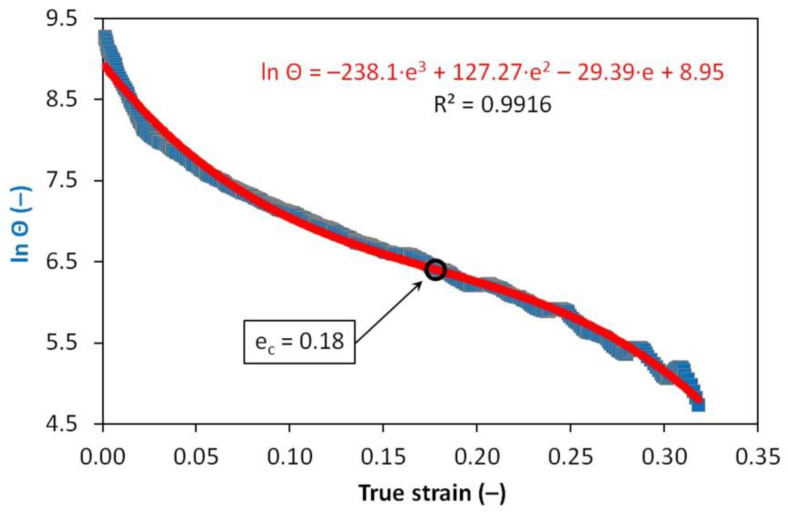
Determination of strain *e_c_* for deformation temperature of 880 °C.

**Figure 4 materials-13-05116-f004:**
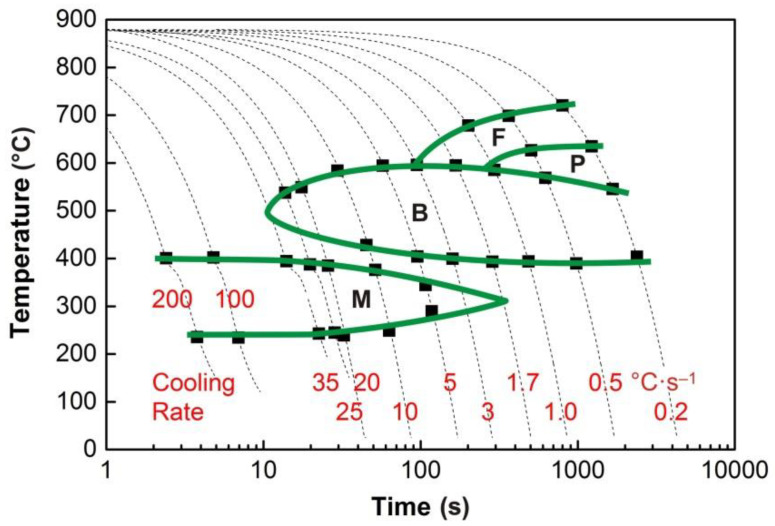
Continuous cooling transformation (CCT) diagram (*e* = 0) for the austenitization temperature of 880 °C.

**Figure 5 materials-13-05116-f005:**
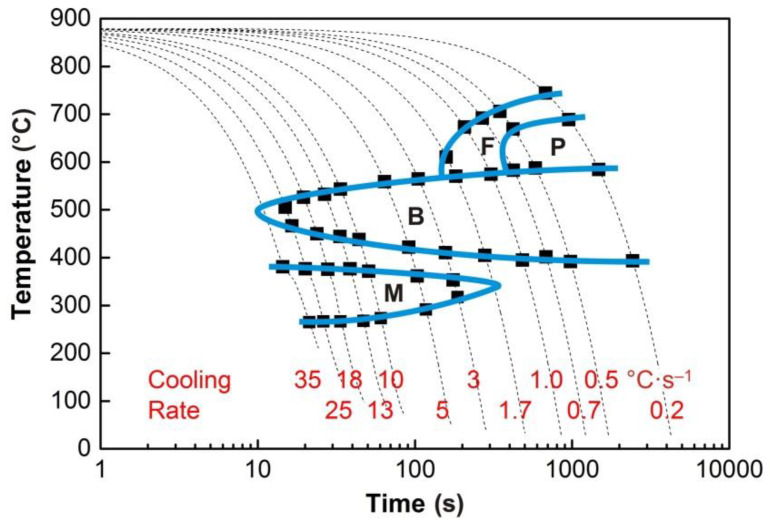
Deformation continuous cooling transformation (DCCT) diagram (*e* = 0.35) for the austenitization temperature of 880 °C.

**Figure 6 materials-13-05116-f006:**
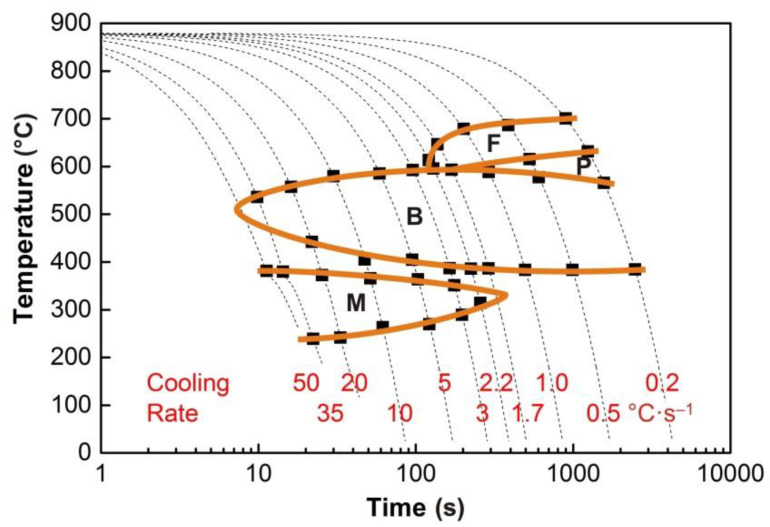
DCCT diagram (*e* = 1.0) for the austenitization temperature of 880 °C.

**Figure 7 materials-13-05116-f007:**
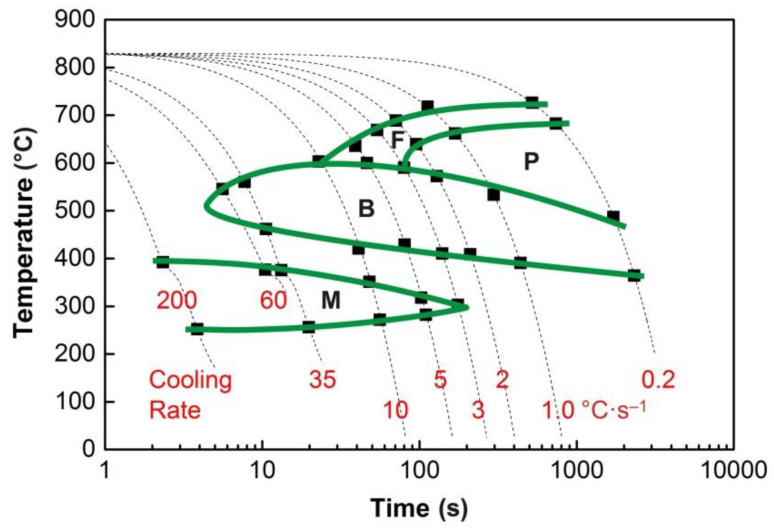
CCT diagram (*e* = 0) for the austenitization temperature of 830 °C.

**Figure 8 materials-13-05116-f008:**
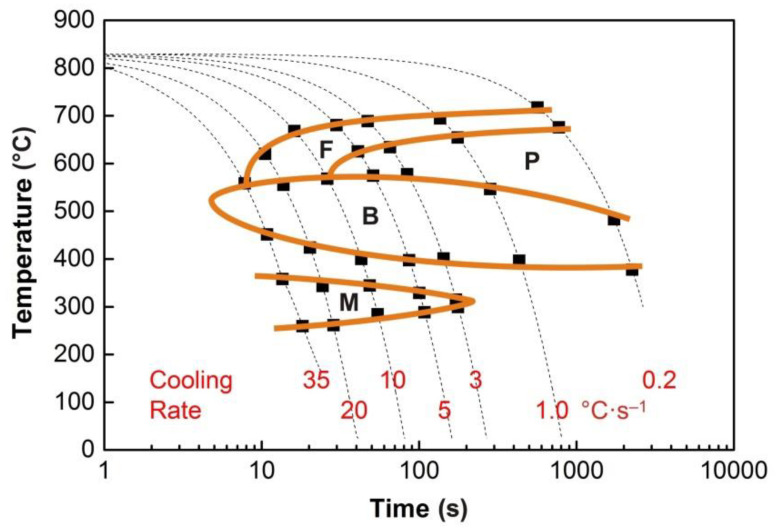
DCCT diagram (*e* = 1.0) for the austenitization temperature of 830 °C.

**Figure 9 materials-13-05116-f009:**
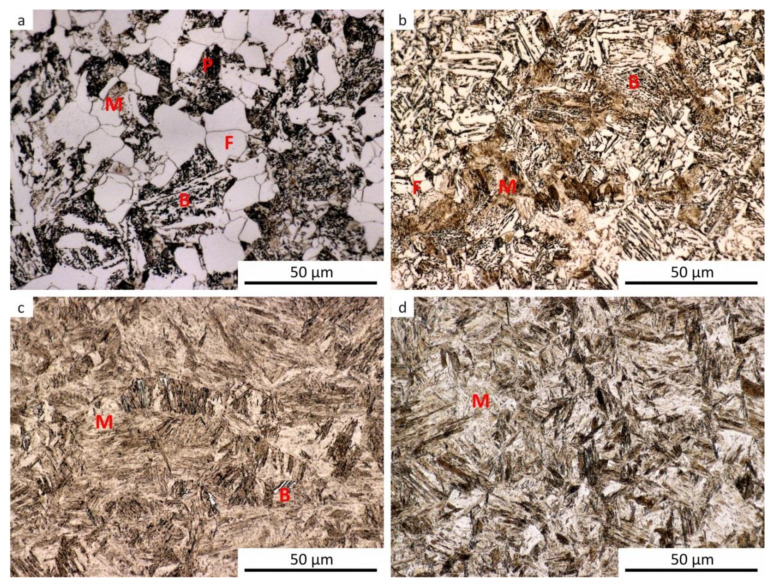
Microstructure of the samples after dilatometry—austenitization at 880 °C, undeformed. (**a**) Cooling rate of 0.2 °C·s^−1^; (**b**) cooling rate of 3 °C·s^−1^; (**c**) cooling rate of 35 °C·s^−1^; (**d**) cooling rate of 100 °C·s^−1^.

**Figure 10 materials-13-05116-f010:**
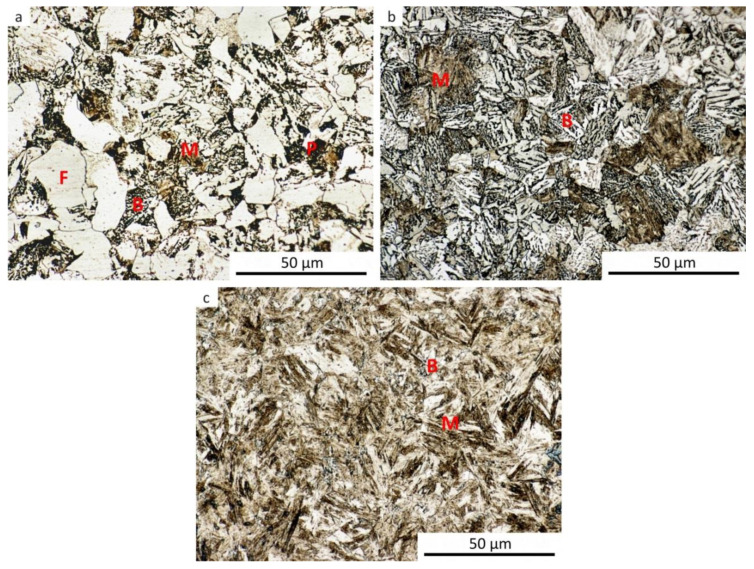
Microstructure of the samples after dilatometry—austenitization at 880 °C, strain *e* = 0.35. (**a**) Cooling rate of 0.2 °C·s^−1^; (**b**) cooling rate of 3 °C·s^−1^; (**c**) cooling rate of 35 °C·s^−1^.

**Figure 11 materials-13-05116-f011:**
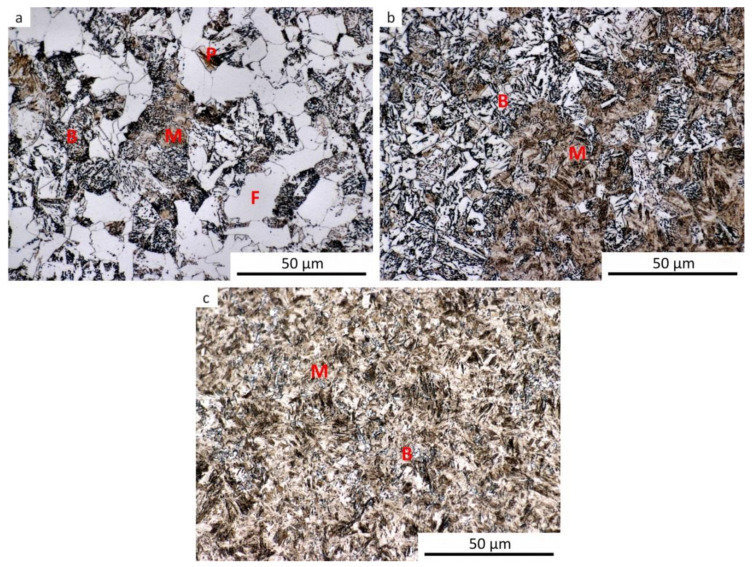
Microstructure of the samples after dilatometry—austenitization at 880 °C, strain *e* = 1.0. (**a**) Cooling rate of 0.2 °C·s^−1^; (**b**) cooling rate of 3 °C·s^−1^; (**c**) cooling rate of 35 °C·s^−1^.

**Figure 12 materials-13-05116-f012:**
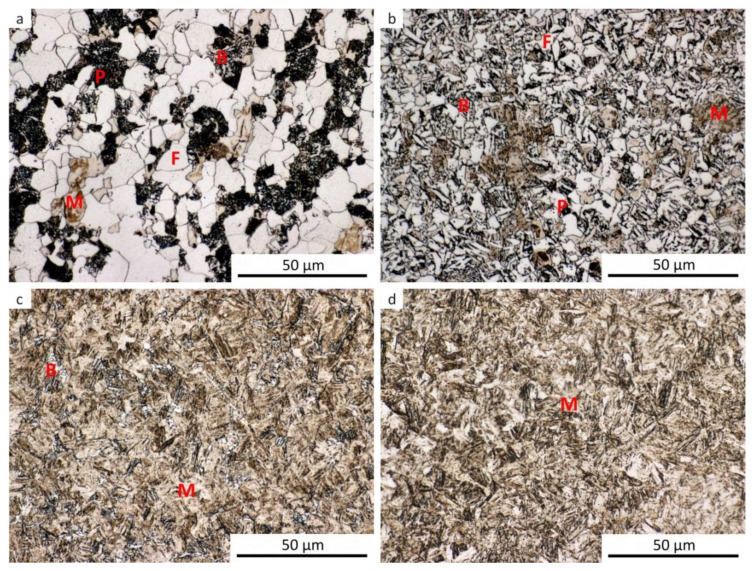
Microstructure of the samples after dilatometry—austenitization at 830 °C, undeformed. (**a**) Cooling rate of 0.2 °C·s^−1^; (**b**) cooling rate of 3 °C·s^−1^; (**c**) cooling rate of 35 °C·s^−1^; (**d**) cooling rate of 100 °C·s^−1^.

**Figure 13 materials-13-05116-f013:**
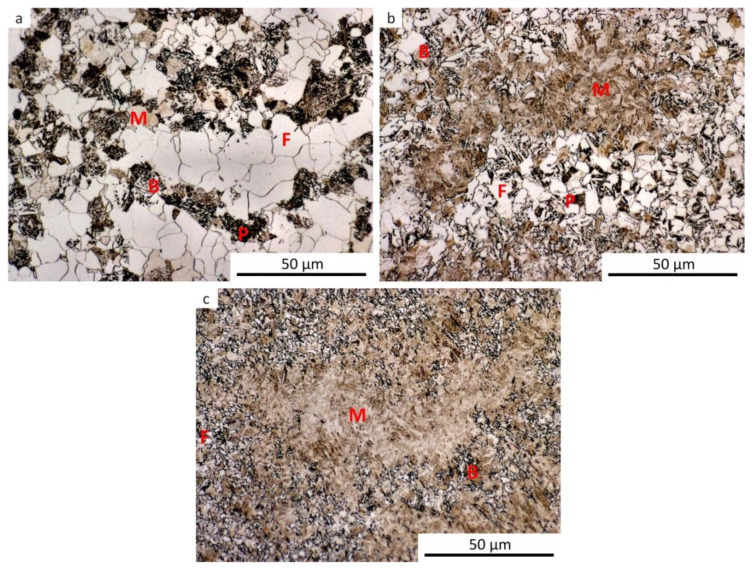
Microstructure of the samples after dilatometry—austenitization at 830 °C, strain *e* = 1.0. (**a**) Cooling rate of 0.2 °C·s^−1^; (**b**) cooling rate of 3 °C·s^−1^; (**c**) cooling rate of 35 °C·s^−1^.

**Figure 14 materials-13-05116-f014:**
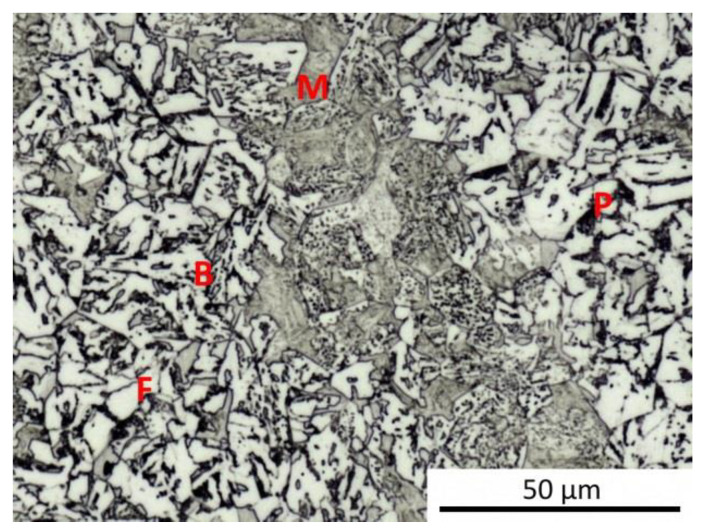
Structure components in the initial hot-rolled state.

**Figure 15 materials-13-05116-f015:**
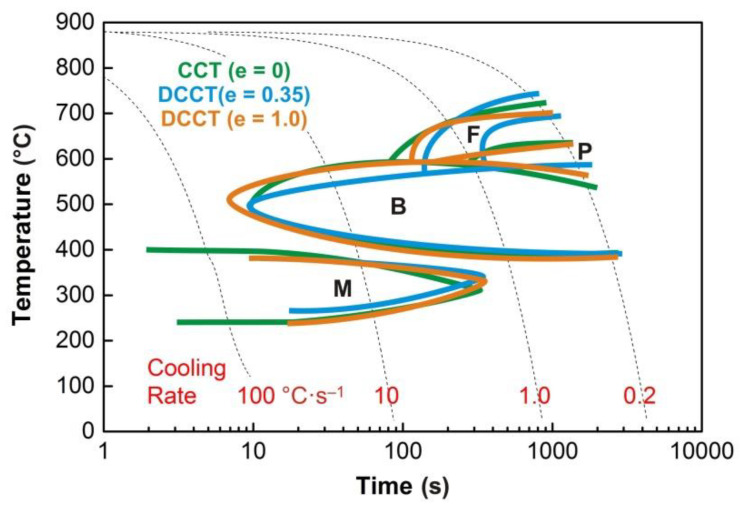
Comparison of the transformation diagrams for the austenitization temperature of 880 °C.

**Figure 16 materials-13-05116-f016:**
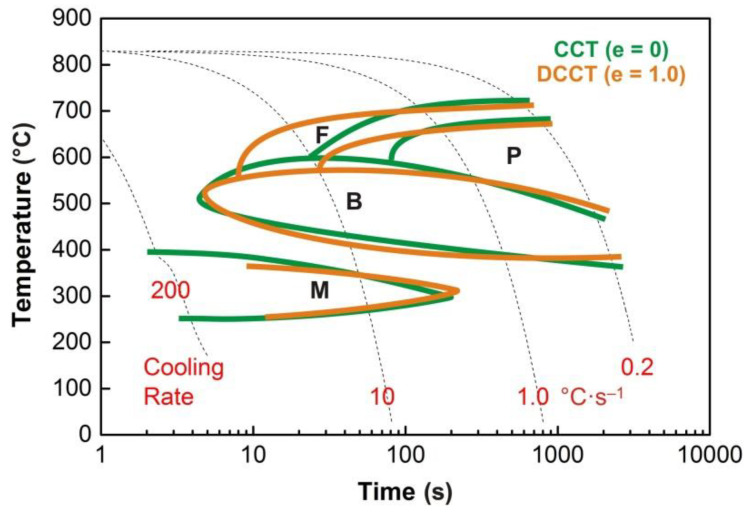
Comparison of the transformation diagrams for the austenitization temperature of 830 °C.

**Figure 17 materials-13-05116-f017:**
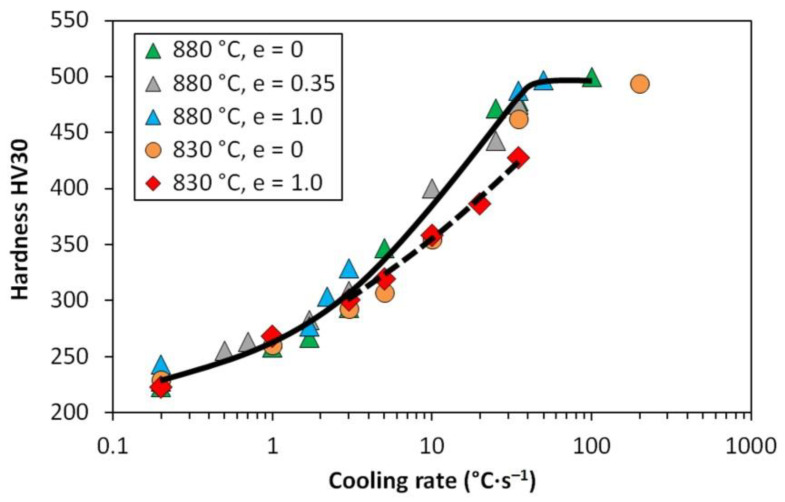
Hardness as a function of experimental conditions and cooling rate.

**Figure 18 materials-13-05116-f018:**
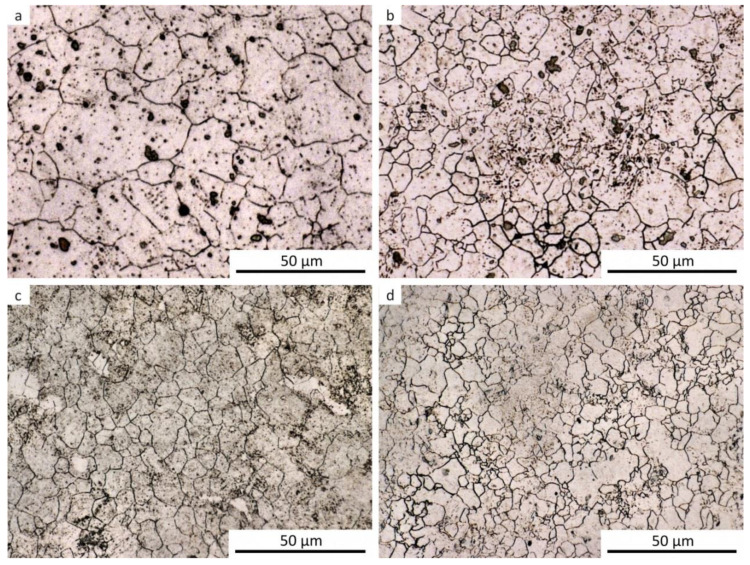
Prior austenitic grain boundaries in the samples cooled at a rate of 10 °C·s^−1^. (**a**) *T_A_* = 880 °C, undeformed; (**b**) *T_A_* = 880 °C, strain *e* = 1.0; (**c**) *T_A_* = 830 °C, undeformed; (**d**) *T_A_* = 830 °C, strain *e* = 1.0.

**Figure 19 materials-13-05116-f019:**
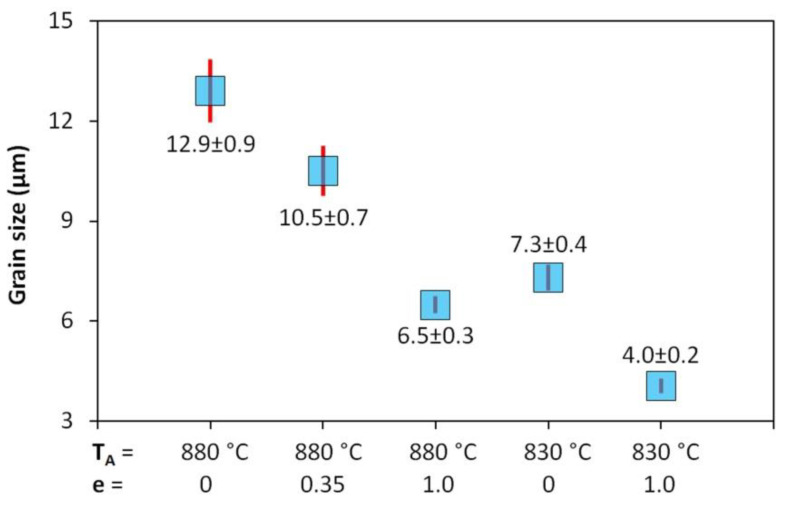
Influence of the austenitization and deformation parameters on the size of prior austenitic grains.

**Figure 20 materials-13-05116-f020:**
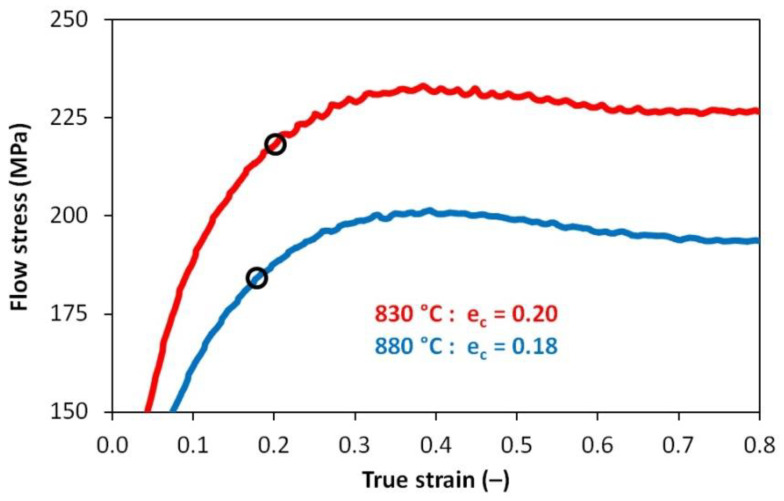
Experimentally determined stress–strain curves and strain *e_c_* values necessary to initiate dynamic recrystallization—detail of the peak stress area.

**Table 1 materials-13-05116-t001:** Chemical composition of the investigated steel in wt.%.

C	Mn	Si	P	S	Ni	Cr	Mo	Al	N
0.23	1.17	0.16	0.015	0.006	0.79	0.44	0.22	0.034	0.0056

**Table 2 materials-13-05116-t002:** Phase composition of the selected samples after dilatometry.

Cooling	Austenitization Temperature/True Strain				
Rate	*T_A_* = 880 °C			*T_A_* = 830 °C	
(°C·s^−1^)	*e* = 0	*e* = 0.35	*e* = 1.0	*e* = 0	*e* = 1.0
0.2	F + P + B	F + P + B	F + B + P	F + P + B	F + B + P
0.5	–	B + F + (P)	–	–	–
0.7	–	B + F + (P)	–	–	–
1.0	F + B + (P)	–	F + B + (P)	F + B + (P)	F + B + (P)
1.7	B + M + (F)	B + M + [F]	B + M + (F) + [P]	–	–
2.2	–	–	B + M + [F]	–	–
3.0	B + M + [F]	B + M	B + M	B + M + F + [P]	B + M + F + (P)
5.0	M + B	–	–	B + M + [F]	M + B + F + [P]
10	–	M + B	–	B + M + [F]	M + B + (F)
20	–	–	–	–	M + B + [F]
25	M + (B)	M + (B)	–	–	–
35	M + (B)	M + (B)	M + (B)	M + (B)	M + (B) + [F]
50	–	–	M + [B]	–	–
100	M	–	–	–	–
200	–	–	–	M	–

**Table 3 materials-13-05116-t003:** Hardness HV30 of the selected samples after dilatometry.

Cooling	Austenitization Temperature/True Strain				
Rate	*T_A_* = 880 °C			*T_A_* = 830 °C	
(°C·s^−1^)	*e* = 0	*e* = 0.35	*e* = 1.0	*e* = 0	*e* = 1.0
0.2	223	228	243	229	223
0.5	–	255	–	–	–
0.7	–	263	–	–	–
1.0	258	–	266	260	268
1.7	267	283	277	–	–
2.2	–	–	303	–	–
3.0	294	309	328	293	300
5.0	347	–	–	308	319
10	–	400	–	355	358
20	–	–	–	–	387
25	472	443	–	–	–
35	478	475	487	462	428
50	–	–	497	–	–
100	500	–	–	–	–
200	–	–	–	494	–

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
