# Peer review of "Effects of Austenitization Temperature and Pre-Deformation on CCT Diagrams of 23MnNiCrMo5-3 Steel"

_materials, 2020, doi:10.3390/ma13225116_

Round 1

Reviewer 1 Report

The manuscript made valuable contribution by examining the effects of austenitization temperature and pre-deformation on CCT diagrams of 23MnNiCrMo5-3 steel through designed experiments. The manuscript is recommended to publish, with a couple of comments below for revision consideration:

  • Line 2: "Effect" should be "Effects"?
  • Line 69: "thesis" could be replaced by "work"?
  • Line 114: please rewrite "These documents the comparison of.." as no verb is available.
  • Will be the future work planed for determining the parameters A, B, Q in Eqs. (1) and (2) by varying strain rate?

Author Response

Thank you for the valuable comments. All of them have been reflected in the revised version of manuscript. All changes are marked in yellow.

The manuscript made valuable contribution by examining the effects of austenitization temperature and pre-deformation on CCT diagrams of 23MnNiCrMo5-3 steel through designed experiments. The manuscript is recommended to publish, with a couple of comments below for revision consideration:

Line 2: "Effect" should be "Effects"?

Good idea – the plural better describes the content of the article.

Line 69: "thesis" could be replaced by "work"?

The word “thesis” was replaced by the preferable “conclusion”.

Line 114: please rewrite "These documents the comparison of.." as no verb is available.

I apologize for the grammatical error. In fact, the word “documents” was intended as a verb. The sentence has been reworded: “This is evidenced by the comparison of the CCT and DCCT diagrams ...”

Will be the future work planed for determining the parameters A, B, Q in Eqs. (1) and (2) by varying strain rate?

So far, we have only the results of 3 uniaxial compression tests, which is of course not enough to mathematically describe these parameters. We intend to supplement the experiment with testing at different temperatures and strain rates, and focus on the effect of the prior austenitic grain size on the kinetics of DRX. We also have interesting experience with the calculation of the activation energy Q based on the stress corresponding to the strain ep, ess and / or ec (including a comparison of the Q values thus obtained).

Reviewer 2 Report

This paper is a correct work. It describes the effect of pre-deformation and the austenization temperature to the CCT and DCCT diagrams. The results are not very surprising, but their interpretation is very correct. I suggest to publish the paper.

Author Response

This paper is a correct work. It describes the effect of pre-deformation and the austenization temperature to the CCT and DCCT diagrams. The results are not very surprising, but their interpretation is very correct. I suggest to publish the paper.

Thank you very much for the exceptionally generous evaluation of our work.

Reviewer 3 Report

The paper investigated the effect of austenitization temperature and pre-deformation on the CCT Diagrams of  23MnNiCrMo5-3 Steel. The experimental processes were described very detailed, but the author did not give more novelty discovery in this paper. I thought this paper should be made major revision.

  1. The author should give some explanation on the application for  23MnNiCrMo5-3 Steel. Furthermore, why did the author select this steel to test?
  2. The author should give the microstructural feature of un-deformed and no cooling rate to compare.
  3. Why did the author select the Austenitization temperature of 880 oC and 830oC?
  4. In figure 19, why did the two curve start from the original point? The author should give some scientific explanation.  

Author Response

Thank you for the inspiring comments. All of them have been reflected in the revised version of manuscript. All changes are marked in yellow.

The paper investigated the effect of austenitization temperature and pre-deformation on the CCT Diagrams of 23MnNiCrMo5-3 Steel. The experimental processes were described very detailed, but the author did not give more novelty discovery in this paper. I thought this paper should be made major revision.

The novelty of the article is that the combined effect of austenitization/deformation temperature and strain value on the CCT of 23MnNiCrMo5-3 steel has been thoroughly researched; pre-deformation was chosen in an unusually wide range of 0 – 1.0. We have tried to state this in the first paragraph of Chapter 2.

1.The author should give some explanation on the application for 23MnNiCrMo5-3 Steel. Furthermore, why did the author select this steel to test?

The information in Chapter 2 was enriched:

“Steel 23MnNiCrMo5-3 is currently one of the most required materials for the production of chains, in terms of price-quality ratio. After cooling from the finish-rolling temperature, it is suitable to achieve the highest possible bainite content in its microstructure. According to operational experience, bainite appears to be the most suitable structural component for annealing, which precedes the actual production of chains. This steel can be used for the welded round link chains and various components for chain hoists, chain conveyors in the mining industry, etc. The final properties of these products are then affected by quenching and tempering.”

2.The author should give the microstructural feature of un-deformed and no cooling rate to compare.

Perhaps I understood correctly that the comment concerns the initial hot-rolled state of the tested material. The micrograph in new Figure 14 and a comparative description of the structure were added in Chapter 3.2:

Figure 14. Structure components in the initial hot-rolled state.

“For comparison, a metallographic analysis of the initial hot-rolled structure of the investigated steel was also performed (see Figure 14). Due to its phase composition, this microstructure is close to structures obtained after dilatometric testing from the lower austenitization temperature and using the cooling rate of about 3 °C·s−1. The hot-rolled microstructure appears to be somewhat coarser.”

Subsequently, the original Figures 14 - 19 had to be renumbered.

If the term “no cooling rate” means the quasi-equilibrium conditions of testing, the extremely low cooling rates were not used because it does not matter from a practical point of view (at the given rolling mill).  However, Chapter 3.2 has at least been supplemented by the following comment:

“In case of the cooling rate minimization and quasi-equilibrium conditions achievement in the dilatometry of the investigated steel, the fraction of ferrite and perlite would further increase at the expense of the bainite and martensite content. However, due to the existence of microsegregations, some trace occurrence of martensite cannot be excluded even under these conditions.”

3.Why did the author select the Austenitization temperature of 880 oC and 830oC?

The text in Chapter 2 was supplemented:

ad TA = 880 °C

“Based on the experimentally obtained value Ac3 = 801 °C and the finish-rolling temperature commonly used on the continuous wire mill in TŘINECKÉ ŽELEZÁRNY a.s., the austenitization temperature TA = 880 °C was used in the first stage of the dilatometric experiments. This value corresponds to the requirements of DIN 17115 German standard for the quenching temperature of the studied steel, which is 870 °C to 890 °C.”

ad TA = 830 °C

“This value is based on the operating limits of the relevant rolling mill and is particularly interesting in terms of the possible refinement of the prior austenitic grains.”

4.In figure 19, why did the two curve start from the original point? The author should give some scientific explanation.

Apparently both curves do not start from point [0;0], which would of course be difficult to explain. However, this is given by the scale used on the vertical axis (from 150 to 250 MPa). It has been already mentioned in the text: “Graph in Figure 20 plots parts of the hot flow curves around the stress peak ...”. Moreover, the figure heading has been clarified: “Figure 20. Experimentally determined stress-strain curves and strain ec values necessary to initiate dynamic recrystallization – detail of the peak stress area.”

Round 2

Reviewer 3 Report

All comments had modified. The paper could be published as this revised form.